# Why and How Auxiliary Tasks Improve JEPA Representations

**Jiacan Yu**
Johns Hopkins University
jyu197@jh.edu

**Siyi Chen**
Johns Hopkins University
schen357@jhu.edu

**Mingrui Liu**
Northwestern University
mingruiliu2025@u.northwestern.edu

**Nono Horiuchi**
University of Rochester
nhoriuch@u.rochester.edu

**Vladimir Braverman**
Johns Hopkins University
vova@cs.jhu.edu

**Zicheng Xu**
Johns Hopkins University
zxu161@jh.edu

**Dan Haramati**
Brown University
dan_haramati@brown.edu

**Randall Balestriero**
Brown University
randall_balestriero@brown.edu

## Abstract

Joint-Embedding Predictive Architecture (JEPA) is increasingly used for visual representation learning and as a component in model-based RL, but its behavior remains poorly understood. We provide a theoretical characterization of a simple, practical JEPA variant that has an auxiliary regression head trained jointly with latent dynamics. We prove a *No Unhealthy Representation Collapse* theorem: in deterministic MDPs, if training drives both the latent-transition consistency loss and the auxiliary regression loss to zero, then any pair of non-equivalent observations, i.e., those that do not have the same transition dynamics or auxiliary value, must map to distinct latent representations. Thus, the auxiliary task anchors which distinctions the representation must preserve. Controlled ablations in a counting environment corroborate the theory and show that training the JEPA model jointly with the auxiliary head generates a richer representation than training them separately. Our work indicates a path to improve JEPA encoders: training them with an auxiliary function that, together with the transition dynamics, encodes the right equivalence relations.

## 1 Introduction

Joint-Embedding Predictive Architecture (JEPA) has become a go-to recipe for image/video representation learning [1, 2] and is increasingly used in model-based Reinforcement Learning (RL) and planning [3, 4, 5, 6, 7]. Yet its success is not "out-of-the-box": practitioners report brittleness and representation collapse unless carefully tuned [8, 9]. What is missing is a theory that explains *which* knobs matter and *why*.

Previous SSL theories only connect methods to each other [10, 11] or provide some guarantees in infinite or nonparametric regime [12, 13, 14]. We provide theoretical statements that hold in realistic finite-data regime with the JEPA loss being used in practice. We consider a minimal, practical variant

Preprint.

where a JEPA model and an auxiliary neural network (Fig. 1) learn consistent latent dynamics and fit a function of observations, i.e. auxiliary function. The key message is simple: the auxiliary task is not a heuristic—it determines the information the representation must preserve. We formalize this via a *No Unhealthy Representation Collapse* theorem (Thm. 1): in deterministic MDPs, if the dynamics-consistency and auxiliary losses reach zero, then any two observations that have different transition dynamics or auxiliary values receive different latent representations. Hence the auxiliary choice controls the type of information encoded in the representation space. This offers a practical lever: improve JEPA encoders *via the auxiliary* rather than ad-hoc architecture tweaks.

We conduct experiments in a counting environment (Sec. 2.3), where the observations are images containing different numbers of objects and actions are adding or removing an object. We find that the learned latent space forms distinct clusters for observations containing different numbers of objects, matching the theory's prediction that there will be one non-collapsible class per object count. Decoders trained without backpropagating into the encoder cannot recover shape, color, or position, showing the encoder's strong capability of abstraction. Our results show that the auxiliary task guides the encoder to distinguish non-equivalent observations. Therefore, JEPA encoders can be improved by choosing auxiliary tasks that, when combined with the transition dynamics, encode helpful equivalence relations.

## 2 Theoretical Characterization of JEPA with Auxiliary Tasks

This section first introduces the model and training objective we consider (Sec. 2.1), followed by a theoretical characterization that formalizes the notion of non-equivalent observations and establishes our main theorem that shows non-equivalent observations will not be collapsed (Thm. 1). We then present experiments in a counting environment, validating the theory through clustering analysis and visualization of the learned latent space.

### 2.1 Setup

Consider a deterministic Markov decision process (MDP) $\mathcal{M} = (\mathcal{O}, \mathcal{A}, \mu, f, r)$ [16], where $\mathcal{O}$ is the observation space (finite in practice due to digital discretization), $\mathcal{A}$ is the action space, $\mu \in \mathcal{P}(\mathcal{O})$ is the initial observation distribution, $f : \mathcal{O} \times \mathcal{A} \to \mathcal{O}$ is the transition dynamics, and $r : \mathcal{O} \to \mathbb{R}$ is the reward function.

Consider a JEPA model with an auxiliary network on top of the encoder, as shown in Fig. 1: a neural network $P_\theta$ regresses to an auxiliary function $p$, and there is no stop gradient in JEPA's latent dynamics loss. $P_\theta$, $E_\phi$, and $T_\psi$ are trained jointly by minimizing the latent transition loss and the auxiliary loss: $\mathcal{L}(\theta, \phi, \psi) = \mathcal{L}_{dyn} + c_p \mathcal{L}_p$, where $c_p$ is a hyperparameter controlling the weight of the auxiliary loss. The latent transition loss is: $\mathbb{E}_{(o_t, a_t, o_{t+1})} ||T_\psi(E_\phi(o_t), a_t) - E_\phi(o_{t+1})||^2$. The auxiliary loss $\mathcal{L}_p$ is a loss that measures the difference between the output of $P_\theta(E_\phi(o))$ and $p$.

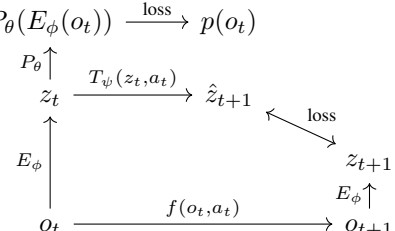

Figure 1: Architecture of P-JEPA. The pentagon is the JEPA core: $E_\phi$ is the encoder; $T_\psi$ is the latent transition model. $P_\theta(z_t)$ regresses to an auxiliary function of observations $p$. $p$ can be the reward $r$ or a randomly initialized neural network; see Sec. 3. $E_\phi$ is updated by both the dynamics loss and the auxiliary loss; no target/EMA [15] encoder is used.

### 2.2 Theory

In the RL literature, equivalence between observations is captured by bisimulation, i.e., having the same reward and transition dynamics [17, 18], but an MDP may admit many bisimulations. We consider the largest one, which includes all equivalent pairs of observations, and replace the reward with an auxiliary function. For proof clarity, we adopt an apartness-based definition (Def. 1): define a monotone operator whose least fixed point collects pairs distinguishable now or after some time steps; its complement is the largest bisimulation [19]. We then show that P-JEPA cannot collapse non-equivalent observations, since it must fit the auxiliary function and maintain consistent latent dynamics.

**Definition 1** (Largest bisimulation). *Let $\mathcal{M}$ be a deterministic MDP, $p$ be a function of observations, and $R \subseteq \mathcal{O}^2$ be a relationship in the observation space. Define the operator $\mathcal{F}(R) := \{(o, o') \in \mathcal{O}^2 : p(o) \neq p(o')\} \cup \{(o, o') \in \mathcal{O}^2 : \exists a \in \mathcal{A} \text{ with } (f(o, a), f(o', a)) \in R\}$. Start from $R^{(0)} = \varnothing$, and iterate $R^{(t+1)} = \mathcal{F}(R^{(t)})$. This process collects pairs of observations distinguishable immediately or after the same sequence of actions. Because $\mathcal{O}^2$ is finite in practice, $\mathcal{F}(R)$ stabilizes after finitely many steps at $R^\star = \mathcal{F}(R^\star)$. $R^\star$ is the least fixed point of $\mathcal{F}$. Define $B^\star := (\mathcal{O} \times \mathcal{O}) \setminus R^\star$. We call $B^\star$ the* largest bisimulation *over $\mathcal{M}$ and $p$.*

**Theorem 1** (No Unhealthy Representation Collapse). *Let $\mathcal{M}$ be a deterministic MDP, $p$ be a function of observations, and a P-JEPA model be well-trained: $T_\psi(E_\phi(o), a) = E_\phi(f(o, a))$ and $P_\theta(E_\phi(o)) = p(o)$ for all $o$ and $a$. Then any pair of observations that is not in the largest bisimulation over $\mathcal{M}$ and $p$ does not collapse: $o_i \not\equiv_{B^\star} o_j \implies E_\phi(o_i) \neq E_\phi(o_j)$.*

Proof in Appx. B. The finite data version of this theorem is in Appx. B.1.

### 2.3 How Auxiliary Tasks Affect Learned Representation

We design a counting environment with $64{\times}64$ RGB observations containing $k \in \{0, \dots, 8\}$ objects. When $k = 0$, an observation is a completely dark image. At episode start, a shape (triangle/disk/square/bar) and color are sampled and fixed. Example observations are shown in the third column of Fig. 2. The actions are increasing or decreasing $k$ by one. Positions of objects are resampled each step. Reward is $1$ iff the count equals a fixed $n$, else $0$. We train our P-JEPA model on a dataset collected by a random policy. We set $n = 4$ in our experiment. The auxiliary task is regressing to the reward.

One may think the JEPA model will collapse the representation to two clusters, because the reward only has two different values. However, according to Def. 1, there are 9 non-bisimilar sets, one per object count. Therefore, Thm. 1 predicts that the observations will be mapped to at least 9 distinct representations (proof in Appx. C). As observed in the top row of Fig. 2, Principal Component Analysis (PCA) [20] visualization of 256 encoding vectors indeed shows nine clusters; pairwise distances within counts are smaller than those across counts, indicating separation of clusters; a decoder trained in parallel *without* backpropagating into the encoder fails to reconstruct shape, color, and positions, indicating that the model is able to abstract away redundant information.

During training, we observe that the encoder sometimes diffuses compact clusters into large blobs, but this does not contradict our theory. Our guarantees are *one-sided*: under perfect training, pairs that are *not* equivalent in the largest bisimulation cannot collapse, but the theory does *not* require bisimilar observations to merge to a single encoding vector. Future studies can look into ways to stabilize collapse within bisimilar classes. In this work, all plots are obtained when the clusters are the most compact. The compactness is measured by nearest-centroid classification accuracy, where each centroid is computed from embeddings with a given object count, and an observation is classified correctly if its true count matches that of its nearest centroid.

We then set the auxiliary function to a fixed random 256-D linear mapping. This makes almost all pairs of observations non-bisimilar, which should prevent most representation collapse. Indeed, as observed from the middle row of Fig. 2, the heatmap shows that embeddings are separated, though not organized by count. The decoder is able to recover the position information and part of the color and shape information, indicating that the encoder partially preserves these factors rather than collapsing them.

Consider the two training losses separately: reward loss alone yields only coarse separation, as observed from the bottom row of Fig. 2; latent transition loss alone leads to complete collapse into a single compact cluster [21], whereas combining them in P-JEPA produces nine separated clusters, showing that our model learns a richer representation.

## 3 Conclusion

**A Knowledge Discovery View of JEPA+Auxiliary Tasks.** A possible interpretation is that the model is trained to discover knowledge from the environment. The learned piece of knowledge is the triple $\mathcal{K} := (E_\phi, T_\psi(z, a), P_\theta(z))$ that explains a user-specified phenomenon. Here (i) $E_\phi$ abstracts observations into representations, (ii) $T_\psi$ enforces transition consistency in latent space, and (iii) $P_\theta$

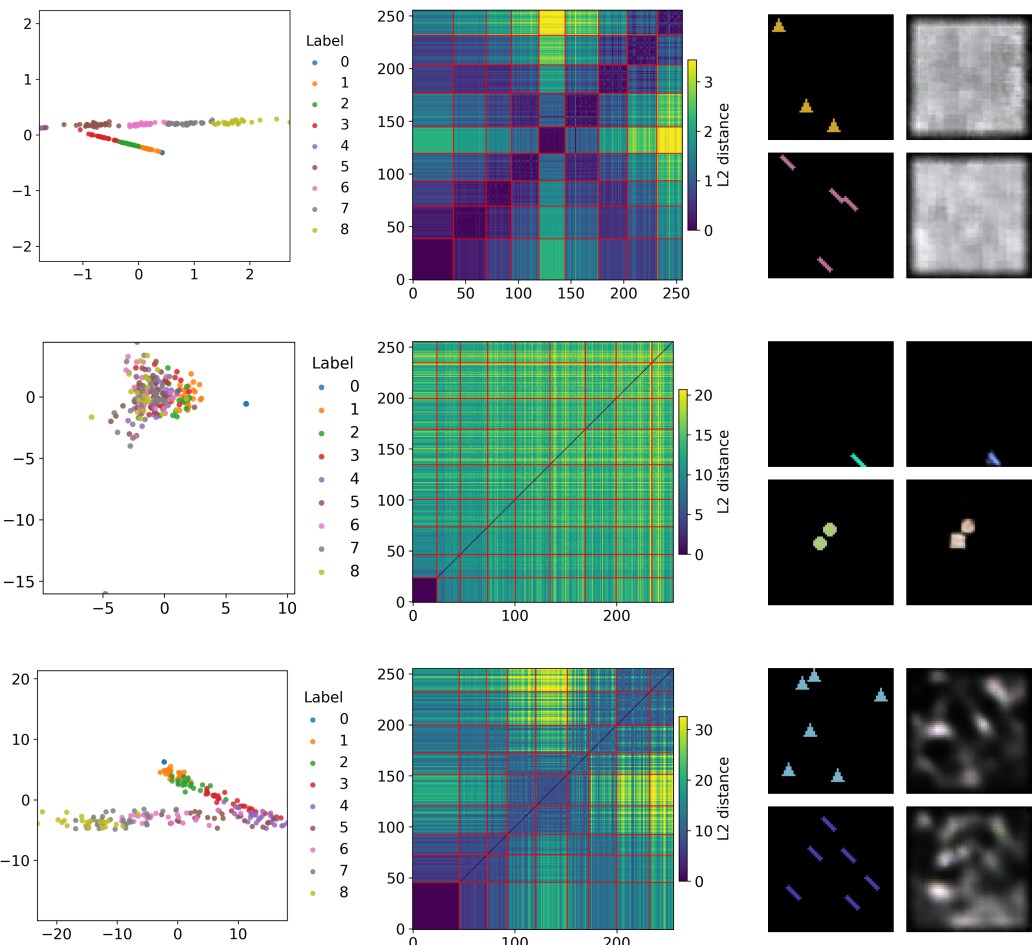

Figure 2: **Each row**: *left*—PCA of embeddings of 256 randomly chosen observations; different colors correspond to different object counts; *middle*— pairwise $\ell_2$ distances between the same 256 embeddings, with samples sorted by object count and red grid lines marking count boundaries; *right*—example observations (left of each pair) and decoder outputs (right, normalized for better contrast). **Top row**: P-JEPA with reward auxiliary: PCA shows nine clear clusters; the diagonal blocks in the heatmap are darker than off-diagonal blocks, indicating separation according to object count; reconstructions discard shape/color/position. **Middle row**: P-JEPA with 256-dimensional random auxiliary: no count structure; in the heatmap, the diagonal blocks are as bright as off-diagonal blocks, showing distances within the same object count are comparable to those across different counts; decoder recovers position and partial color/shape information. **Bottom row**: Encoder receives gradients only from reward loss: representation space shows only coarse separation; the heatmap exhibits only coarse block structure, roughly grouping counts into three sets (0–2, 3–5, 6–8); decoder cannot recover color/shape/position information.

predicts the phenomenon value. Knowledge discovery requires a dataset of observed transitions and phenomenon values $\mathcal{D} = \{(o, a, f(o, a), p(o))\}$. The objective of knowledge discovery decomposes into two parts: (1) make $E_\phi, T_\psi$ consistent with dynamics; (2) fit $P_\theta(E_\phi(o))$ to the phenomenon. Crucially, the loss does not require maximization of the phenomenon function during training; actions can be random. Control can be done *after* learning $\mathcal{K}$ by planning [22] over the learned dynamics using model predictive control. Our focus in this work is characterizing what knowledge can be learned given an environment and a phenomenon function. In vanilla JEPA, the task is "explaining

nothing". The knowledge that explains nothing is only required to be consistent, and the easiest way for $\mathcal{K}$ to be consistent is complete representation collapse.

**Improving JEPA encoders.** Our theory suggests a way to improve JEPA encoders: introduce an auxiliary function that represents the phenomenon that the representation should explain. The auxiliary function and the transition dynamics define an equivalence relation over observations (Def. 1), guiding the encoder to collapse only within equivalence classes while preserving distinctions across non-equivalent ones. Thus, the encoder can discard irrelevant variation while preserving distinctions that matter for the task. In RL, natural choices of the auxiliary function include the reward or Q-function, as used in TD-MPC2 [4]. Our results thus provide theoretical grounding for why such designs are effective.

**Conclusion.** We gave a simple, actionable characterization of JEPA with an auxiliary regression head: the auxiliary target anchors which distinctions the encoder must preserve. Formally, under perfect training in deterministic MDPs, non–bisimilar observations cannot map to the same encoding vector. Experiments in a counting environment match these predictions and show that redundant factors are discarded. Practically, one can improve JEPA encoders by choosing the auxiliary to match the phenomenon of interest (e.g., reward/$Q$), clarifying why TD-MPC2-style designs are effective.

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

# Appendix

## A   Related Work

Classical works on bisimulation formalize that when two states are behaviorally indistinguishable: having identical rewards and transition dynamics, they can be considered as the same and task-irrelevant variation can be discarded [23, 17]. Building on this principle, DeepMDP links bisimulation to model-based reinforcement learning methods by adding reward and transition prediction as auxiliary objectives [24]. Our model is trained on just these two objectives and our theory reaches a complementary conclusion: pairs of observations that are not in the largest bisimulation cannot be mapped to the same representation. Zhang et al. propose to learn only task-relevant information by shaping latent distances to match bisimulation distances [25]. We operationalize the removal of irrelevant information via neural-collapse [26, 27] under JEPA training rather than directly optimizing a bisimulation loss.

TD-MPC2 [4] implements the training of a JEPA model in Reinforcement Learning (RL) environments with additional policy, reward, and Q networks on top of the JEPA model. They use a stop gradient in the latent transition loss of JEPA. To understand the representation learned by a JEPA-style model, we experiment with a simplified version of TD-MPC2, which we call P-JEPA. Our implementation is based on the code of TD-MPC2 and is available at `https://github.com/jasonyu48/concept_discovery`.

PLDM (Planning with Latent Dynamics Models) [5] studies learning from reward-free offline trajectories by first training a JEPA model and then performing planning in the learned latent space, thus explicitly separating representation/knowledge discovery from control. They demonstrate their method is data efficient and powerful in generalizing to unseen layouts, supporting a workflow in which discovery of environment regularities precedes task-specific control.

Kondapaneni and Perona [28] also study the structure of learned representations in a similar counting environment, but under a different setup. In particular, they do not have a latent dynamics model, and their task is to predict what action was done based on representations from the previous and the current time step. Despite these differences in model architecture and task, they likewise observe clusters corresponding to object counts in the representation space, indicating that this phenomenon arises broadly across settings.

## B   Proof

In RL, equivalence of observations is captured by bisimulation:

**Definition 2** (Bisimulation for deterministic MDP). *Given a deterministic MDP $\mathcal{M}$, an equivalence relation $B$ between observations is a* bisimulation relation *if, for all observations $o_i, o_j \in \mathcal{O}$ that are equivalent under $B$ (denoted $o_i \equiv_B o_j$) the following conditions hold (i) $r(o_i) = r(o_j)$ and (ii) $f(o_i, a) \equiv_B f(o_j, a) \; \forall a \in \mathcal{A}$ [23, 17, 18].*

We replace the reward function by the auxiliary function, and consider the relationship that contains all equivalent pairs of observations:

**Definition 3** (Largest bisimulation). *Let $\mathcal{M}$ be a deterministic MDP, $p$ be a function of observations, and $R \subseteq \mathcal{O}^2$ be a relationship in the observation space. Define the operator $\mathcal{F}(R) := \{(o, o') \in \mathcal{O}^2 : p(o) \neq p(o')\} \cup \{(o, o') \in \mathcal{O}^2 : \exists a \in \mathcal{A} \text{ with } (f(o, a), f(o', a)) \in R\}$. Start from $R^{(0)} = \varnothing$, and iterate $R^{(t+1)} = \mathcal{F}(R^{(t)})$. This process collects pairs of observations distinguishable immediately or after the same sequence of actions. Because $\mathcal{O}^2$ is finite in practice, $\mathcal{F}(R)$ stabilizes after finitely many steps at $R^\star = \mathcal{F}(R^\star)$. $R^\star$ is the least fixed point of $\mathcal{F}$. Define $B^\star := (\mathcal{O} \times \mathcal{O}) \setminus R^\star$. We call $B^\star$ the* largest bisimulation *over $\mathcal{M}$ and $p$.*

Then we show that a well-trained P-JEPA model cannot collapse non-equivalent observations.

**Theorem 2** (No Unhealthy Representation Collapse). *Let $\mathcal{M}$ be a deterministic MDP, and the* P-JEPA *model be well-trained:*

$$T_\psi(E_\phi(o), a) = E_\phi\big(f(o, a)\big) \qquad o \in \mathcal{O}, \; a \in \mathcal{A},$$

$$P_\theta(E_\phi(o)) = p(o), \qquad \forall o \in \mathcal{O}.$$

*Then any pair of observations that is not bisimilar in the largest bisimulation over $\mathcal{M}$ and $p$ does not collapse:*

$$o_i \not\equiv_{B^\star} o_j \implies E_\phi(o_i) \neq E_\phi(o_j),$$

*where $B^\star$ is the largest bisimulation relation over $\mathcal{M}$.*

*Proof.* Let $o_1, o_2 \in \mathcal{O}$ with $o_1 \not\equiv_{B^\star} o_2$. By definition, this means that *either*

A. $p(o_1) \neq p(o_2)$, *or*

B. $p(o_1) = p(o_2)$ but their transition behaviors differ: there exists $a \in \mathcal{A}$ such that $(f(o_1, a), f(o_2, a)) \notin B^\star$.

**Case A.** We argue by contradiction. If $E_\phi(o_1) = E_\phi(o_2)$, then $p(o_1) = P_\theta(E_\phi(o_1)) = P_\theta(E_\phi(o_2)) = p(o_2)$, contradicting with $p(o_1) \neq p(o_2)$. Therefore, $E_\phi(o_1) \neq E_\phi(o_2)$.

**Case B.** Suppose $p(o_1) = p(o_2)$ but $o_1 \not\equiv_{B^\star} o_2$. Since $B^\star$ is the largest bisimulation, $(o_1, o_2) \notin B^\star$ implies $(o_1, o_2) \in R^\star$, where $R^\star$ is the least fixed point of $\mathcal{F}$ used to construct $B$. By the construction of $R^\star$ there exists a minimal $k \geq 1$ and actions $a_1, \ldots, a_k$ such that, writing

$$o_1^{(0)} = o_1, \quad o_2^{(0)} = o_2, \qquad o_i^{(t+1)} = f\big(o_i^{(t)}, a_{t+1}\big) \ (t = 0, \ldots, k-1),$$

we have $p(o_1^{(t)}) = p(o_2^{(t)})$ for all $t < k$ and $p(o_1^{(k)}) \neq p(o_2^{(k)})$. For each $t < k$, well-trainedness gives $T_\psi(E_\phi(o_1^{(t)}), a_{t+1}) = E_\phi(o_1^{(t+1)})$. We prove by backward induction on $t = k, k-1, \ldots, 0$ that $E_\phi(o_1^{(t)}) \neq E_\phi(o_2^{(t)})$.

Base $(t = k)$:

$p(o_1^{(k)}) \neq p(o_2^{(k)})$, we can apply the same argument as Case A. to get $E_\phi(o_1^{(k)}) \neq E_\phi(o_2^{(k)})$.

Inductive step:

Assume $E_\phi(o_1^{(t+1)}) \neq E_\phi(o_2^{(t+1)})$ for some $t < k$ yet $E_\phi(o_1^{(t)}) = E_\phi(o_2^{(t)})$. Then

$$E_\phi(o_1^{(t+1)}) = T_\psi\big(E_\phi(o_1^{(t)}), a_{t+1}\big) = T_\psi\big(E_\phi(o_2^{(t)}), a_{t+1}\big) = E_\phi(o_2^{(t+1)}),$$

contradicting the inductive hypothesis. Hence $E_\phi(o_1^{(t)}) \neq E_\phi(o_2^{(t)})$. In particular $E_\phi(o_1) \neq E_\phi(o_2)$, completing Case B.

$\square$

## B.1 Finite data regime

We analyze the finite data regime in which we do not have access to the ground truth transition or auxiliary function, but only observe a dataset of transitions and auxiliary function values

$$\mathcal{D} \subseteq \mathcal{O} \times \mathcal{A} \times \mathcal{O} \times P, \qquad (o, a, f(o, a), p(o)) \in \mathcal{D},$$

where $P$ is the codomain of the auxiliary function. We assume deterministic transitions: given $o$ and $a$, there can be only one $f(o, a)$. Let

$$\mathcal{O}_D := \left\{ o \in \mathcal{O} \,\middle|\, \exists (o, a, f(o, a), p(o)) \in \mathcal{D} \right\}$$

be the set of observations that appear in $\mathcal{D}$ as sources.

Define the set of *co-observed actions*

$$\mathcal{A}_\cap(x, y) := \left\{ a \in \mathcal{A} \,\middle|\, (x, a, f(x, a), p(x)) \in \mathcal{D} \text{ and } (y, a, f(y, a), p(y)) \in \mathcal{D} \right\}.$$

**Definition 4** (Empirical largest bisimulation). *Define an operator $\mathcal{F}_D$ on $R \subseteq \mathcal{O}_D^2$ by*

$$\mathcal{F}_D(R) := \{(x, y) \in \mathcal{O}_D^2 : p(x) \neq p(y)\}$$
$$\cup \left\{ (x, y) \in \mathcal{O}_D^2 : \exists a \in \mathcal{A}_\cap(x, y) \text{ s.t. } \big(f(x, a), f(y, a)\big) \in R \right\}.$$

*Start from $R^{(0)} = \varnothing$ and iterate $R^{(t+1)} = \mathcal{F}_D(R^{(t)})$. Because $\mathcal{O}_D$ is finite and $\mathcal{F}_D$ is monotone, the chain stabilizes after finitely many steps at the least fixed point $R_D^\star$ with $R_D^\star = \mathcal{F}_D(R_D^\star)$. Set*

$$B_D^\star := (\mathcal{O}_D \times \mathcal{O}_D) \setminus R_D^\star.$$

*We call $B_D^\star$ the* empirical largest bisimulation *over $\mathcal{D}$.*

We say $(E_\phi, T_\psi, P_\theta)$ *is well-trained on $\mathcal{D}$ if*

$$\forall (o, a, f(o,a), p(o)) \in \mathcal{D}: \qquad T_\psi\big(E_\phi(o), a\big) = E_\phi\big(f(o,a)\big) \quad \text{and} \quad P_\theta\big(E_\phi(o)\big) = p(o). \quad (1)$$

**Theorem 3** (Empirical No Unhealthy Representation Collapse)**.** *Suppose $(E_\phi, T_\psi, P_\theta)$ is well-trained on $\mathcal{D}$. Then for all $o_i, o_j \in \mathcal{O}_D$,*

$$(o_i, o_j) \notin B_D^\star \quad \Longrightarrow \quad E_\phi(o_i) \neq E_\phi(o_j).$$

*Equivalently, every pair that the data already certifies as* empirically non-bisimilar *(i.e., in $R_D^\star$) cannot collapse under $E_\phi$.*

*Proof.* Since $(o_i, o_j) \notin B_D^\star$, we have $(o_i, o_j) \in R_D^\star$. Let $R^{(t)}$ be the ascending sequence from Definition 4. We prove by induction on $t$ that $(x, y) \in R_D^\star \Rightarrow E_\phi(x) \neq E_\phi(y)$.

Base:

We want to show $(x, y) \in R^{(1)} \Rightarrow E_\phi(x) \neq E_\phi(y)$. Note that $(x, y) \in R^{(1)}$ iff $p(x) \neq p(y)$. If $E_\phi(x) = E_\phi(y)$, the perfect-fit condition (1) implies $p(x) = P_\theta(E_\phi(x)) = P_\theta(E_\phi(y)) = p(y)$, a contradiction. Hence $E_\phi(x) \neq E_\phi(y)$.

Inductive step:

Take $(x, y) \in R^{(k+1)} \setminus R^{(k)}$. By the successor disagreement clause, there exists $a \in \mathcal{A}_\cap(x, y)$ such that $\big(f(x, a), f(y, a)\big) \in R^{(k)}$. By the inductive hypothesis, $E_\phi(f(x, a)) \neq E_\phi(f(y, a))$. Suppose $E_\phi(x) = E_\phi(y)$. Using the definition of a well-trained model,

$$E_\phi\big(f(x, a)\big) = T_\psi\big(E_\phi(x), a\big) = T_\psi\big(E_\phi(y), a\big) = E_\phi\big(f(y, a)\big),$$

contradicting the inductive hypothesis. Therefore $E_\phi(x) \neq E_\phi(y)$. □

Observations that are not bisimilar in the ground truth can be collapsed if they appear in the dataset only as successors or if the dataset does not cover actions that show they have different transition dynamics, but this is appropriate when the data coverage is not enough. When more data is available, if it is observed that they have different auxiliary values or different transition dynamics, they will be mapped to different representations. Under the knowledge discovery interpretation, this is consistent with the Fallibilism philosophy [29] of Theory of Knowledge: knowledge is fallible, but can be improved after more observations become available.

Observations that cannot collapse when the dataset size is small cannot be collapsed after more data is observed. The reason is that once a pair of observations is added to $R_D^\star$, they are not allowed to collapse, since adding more data will not shrink $R_D^\star$.

## C  Theory's Prediction

Recall our counting environment (§2.3): observations $o$ are $64 \times 64$ images containing $\text{num\_obj}(o) \in \{0, \ldots, 8\}$ objects; actions are $\mathcal{A} = \{\text{inc}, \text{dec}\}$; the auxiliary function is the reward function: $r(o) = \mathbf{1}\{\text{num\_obj}(o) == n\}$, which is an indicator function that shows when $\text{num\_obj}$ is $n \in \{0, \ldots, 8\}$. A shape (triangle/disk/square/bar) and color are sampled in the beginning of an episode and held fixed. The positions of the objects are resampled after each action. Define the 9 subsets

$$G_k := \{o \in \mathcal{O} \mid \text{num\_obj}(o) = k\}, \qquad k = 0, \ldots, 8.$$

The dynamics of the environment can be stated using these subsets: for any $o \in G_k$,

$$f(o, \text{inc}) \in G_{\min\{k+1, 8\}}, \qquad f(o, \text{dec}) \in G_{\max\{k-1, 0\}}.$$

**Proposition 1** (9-way partition is a bisimulation). *Let $B_{cnt} := \bigcup_{k=0}^{8}(G_k \times G_k)$. Then $B_{cnt}$ is a bisimulation.*

*Proof.* Take $(x, y) \in B_{cnt}$. Then $x, y \in G_k$ for some $k$. Consider the rewards: $r(x) = \mathbf{1}\{k = n\} = r(y)$. Consider the dynamics: $f(x, \text{inc}), f(y, \text{inc}) \in G_{\min\{k+1, 8\}}$, $f(x, \text{dec}), f(y, \text{dec}) \in G_{\max\{k-1, 0\}}$, hence the pairs of successors remain in $B_{cnt}$. $\square$

**Proposition 2** (9-way partition is the largest). *We prove 9-way partition is the largest bisimulation over the counting environment.*

*Proof.* Let $G_k = \{o : \#\text{obj}(o) = k\}$ and $p(o) = \mathbf{1}\{\#\text{obj}(o) = n\}$. Let $R^\star$ be the least fixed point of $\mathcal{F}$, as defined in Def.1, and set

$$R_{\neq} := \bigcup_{k \neq \ell}(G_k \times G_\ell).$$

**1)** $R_{\neq} \subseteq R^\star$. Take $(o, o') \in G_k \times G_\ell$ with $k \neq \ell$. Define $d(x) := |\#\text{obj}(x) - n|$, and let $t = \min\{d(o), d(o')\}$. Choose $a^\star = \text{inc}$ if $\#\text{obj}(o) < n$, else $a^\star = \text{dec}$. After $t$ steps of $a^\star$, $o^{(t)}$ is at count $n$ so $p(o^{(t)}) = 1$, while $o'^{(t)}$ is not at $n$, so $p(o'^{(t)}) = 0$. Thus $(o^{(t)}, o'^{(t)}) \in R^{(1)}$. By the successor clause, $(o, o') \in R^\star$. Hence all cross-count pairs lie in $R^\star$.

**2)** $R^\star \subseteq R_{\neq}$. We want to show that at no stage $t$ does a pair from the same $G_k$ appear in $R^{(t)}$.

Base:

$R^{(1)}$ only contains pairs with differing reward values, hence no pair from the same $G_k$ appears in $R^{(1)}$. Therefore $R^{(1)} \subseteq R_{\neq}$.

Inductive step:

If $(x, y) \in G_k \times G_k$, then for any action $f(x, a), f(y, a) \in G_{k'} \times G_{k'}$; by hypothesis this successor is not in $R^{(t)}$, so $(x, y) \notin R^{(t+1)}$. Hence no pair of observations with the same object count is contained in $R^\star$.

Therefore $R^\star = R_{\neq}$ and

$$B^\star = (\mathcal{O}^2) \setminus R^\star = (\mathcal{O}^2) \setminus R_{\neq} = \bigcup_{k=0}^{8}(G_k \times G_k) = B_{cnt}.$$

So the 9-way partition is exactly the largest bisimulation. $\square$

**Corollary 1** (Nine non-collapsible classes). *The largest bisimulation over the counting environment is $B_{cnt}$. The quotient $\mathcal{O}/B_{cnt} = \{G_0, \ldots, G_8\}$. By Thm. 1, any well-trained model cannot map two observations from different $G_k$'s to the same encoding vector.*

# D   Experimental Details

**Environment.** We use the counting environment producing RGB observations of size $3 \times 64 \times 64$. The action space is 1-D continuous in $[-1, 1]$. The sign of the action determines whether the number of objects increases or decreases. The reward is 1 when the number of objects is 4 (success), otherwise it is 0. The environment is episodic with a one-step grace after success. This is because if the episode ends right after success, the model will never see a reward of 1.

**Agent and model.**   - Encoder: 69-layer ResNet-style [30] CNN mapping RGB to a 256-D latent (`latent_dim=256`); pixel preprocessing to $[-0.5, 0.5]$. The deep encoder is to encourage collapse within bisimilar classes, inspired by the "tunnel effect" of deep networks [31]. Our theory does not guarantee collapse within bisimilar classes.

- Dynamics: 2-layer MLP on $[z_t, a_t]$ with hidden dim 512.

- Reward head: 3-layer MLP on $z_t$ with hidden dim 512.

- Decoder: 6-layer convolutional decoder trained with MSE on normalized images $[-1, 1]$; no gradient to encoder.

- Action conditioned reward head, Termination head, Q-function head, and policy prior head are also inherited from the TD-MPC2 implementation, but their gradient flow to the encoder is disabled.

Full model architecture in PyTorch-like notations:

```
Total parameters: 14,866,965
Encoder: 12,260,192
Dynamics: 264,448
Action Conditioned Reward: 448,613
Reward: 448,101
Termination: 396,801
Policy prior: 397,314
Q-functions: 448,613
Decoder: 202,883

Encoder: ModuleDict(
  (rgb): ResNetEncoder(
    (shift): Identity()
    (pre): PixelPreprocess()
    (stem): Sequential(
      (0): Conv2d(3, 32, kernel_size=(7, 7), stride=(2, 2), padding
          =(3, 3), bias=False)
      (1): ReLU()
      (2): MaxPool2d(kernel_size=3, stride=2, padding=1, dilation=1,
          ceil_mode=False)
    )
    (stages): Sequential(
      (0): Sequential(
        (0): _BasicBlock(
          (conv1): Conv2d(32, 32, kernel_size=(3, 3), stride=(1, 1),
              padding=(1, 1), bias=False)
          (relu): ReLU()
          (conv2): Conv2d(32, 32, kernel_size=(3, 3), stride=(1, 1),
              padding=(1, 1), bias=False)
        )
        # ...[7 more _BasicBlock's]...
        )
      )
      (1): Sequential(
        (0): _BasicBlock(
          (conv1): Conv2d(32, 64, kernel_size=(3, 3), stride=(2, 2),
              padding=(1, 1), bias=False)
          (relu): ReLU()
          (conv2): Conv2d(64, 64, kernel_size=(3, 3), stride=(1, 1),
              padding=(1, 1), bias=False)
          (downsample): Conv2d(32, 64, kernel_size=(1, 1), stride=(2,
              2), bias=False)
        )
        # ...[7 more _BasicBlock's]...
        )
      )
      (2): Sequential(
        (0): _BasicBlock(
          (conv1): Conv2d(64, 128, kernel_size=(3, 3), stride=(2, 2),
              padding=(1, 1), bias=False)
          (relu): ReLU()
          (conv2): Conv2d(128, 128, kernel_size=(3, 3), stride=(1, 1),
              padding=(1, 1), bias=False)
          (downsample): Conv2d(64, 128, kernel_size=(1, 1), stride=(2,
              2), bias=False)
        )
        # ...[7 more _BasicBlock's]...
        )
      )
      (3): Sequential(
```

```
      (0): _BasicBlock(
        (conv1): Conv2d(128, 256, kernel_size=(3, 3), stride=(2, 2),
            padding=(1, 1), bias=False)
        (relu): ReLU()
        (conv2): Conv2d(256, 256, kernel_size=(3, 3), stride=(1, 1),
            padding=(1, 1), bias=False)
        (downsample): Conv2d(128, 256, kernel_size=(1, 1), stride
            =(2, 2), bias=False)
      )
      # ...[7 more _BasicBlock's]...
      )
    )
  )
  (avgpool): AdaptiveAvgPool2d(output_size=(1, 1))
  (proj): Linear(in_features=256, out_features=256, bias=True)
  )
)
Dynamics: Sequential(
  (0): NormedLinear(in_features=257, out_features=512, bias=True, act=
      Mish)
  (1): Linear(in_features=512, out_features=256, bias=True)
)
Action Conditioned Reward: Sequential(
  (0): NormedLinear(in_features=257, out_features=512, bias=True, act=
      Mish)
  (1): NormedLinear(in_features=512, out_features=512, bias=True, act=
      Mish)
  (2): Linear(in_features=512, out_features=101, bias=True)
)
Termination: Sequential(
  (0): NormedLinear(in_features=256, out_features=512, bias=True, act=
      Mish)
  (1): NormedLinear(in_features=512, out_features=512, bias=True, act=
      Mish)
  (2): Linear(in_features=512, out_features=1, bias=True)
)
Policy prior: Sequential(
  (0): NormedLinear(in_features=256, out_features=512, bias=True, act=
      Mish)
  (1): NormedLinear(in_features=512, out_features=512, bias=True, act=
      Mish)
  (2): Linear(in_features=512, out_features=2, bias=True)
)
Q-functions: Vectorized 1x Sequential(
  (0): NormedLinear(in_features=257, out_features=512, bias=True,
      dropout=0.01, act=Mish)
  (1): NormedLinear(in_features=512, out_features=512, bias=True, act=
      Mish)
  (2): Linear(in_features=512, out_features=101, bias=True)
)
Reward: Sequential(
  (0): NormedLinear(in_features=256, out_features=512, bias=True, act=
      Mish)
  (1): NormedLinear(in_features=512, out_features=512, bias=True, act=
      Mish)
  (2): Linear(in_features=512, out_features=101, bias=True)
)
```

**Optimization and targets.** We train the model using the Adam optimizer with a base learning rate of $3 \times 10^{-4}$. The encoder parameters use a scaled learning rate of $0.3$ times the base value. Reward and Q-values are transformed using the symlog function and then discretized into two-hot vectors following the TD-MPC2 implementation. Then the reward and Q heads are trained using cross entropy loss. The latent dynamics and the decoder are trained using MSE loss. For the reward-only

experiment, we use a smaller learning rate of $1 \times 10^{-5}$ for all components of the model because the original learning rate cannot make the model converge.

**Data collection and training.**  During data collection, the agent selects actions uniformly at random from the interval $[-1, 1]$. Each sampled action is repeated for four consecutive steps before resampling, which encourages broader exploration of the environment. Transitions are stored in a replay buffer with a capacity of $100,000$, from which mini-batches of size $256$ are drawn for training. The agent interacts with the environment for a total of $300,000$ steps.

