# OpenReview forum: "Why and How Auxiliary Tasks Improve JEPA Representations"
_NeurIPS.cc/2025/Workshop/UniReps — UniReps2025_

### Official Review · Reviewer_TTJu · 2025-09-09

**Confidence:** 4

**Review:**

**Summary**

This work introduces a JEPA loss for state-action-state triplets augmented with an auxiliary task and study the representation learn by such a loss. More precisely, the authors defines an equivalence relations over states and establishes that every pair of states that are not equivalent must have different representations, i.e. should not be collapsed. A simple toy experiment shows that the theory is consistent with observations, at least in this limited setting.

**Review**

I think this paper is very much aligned with the topic of the workshop. It was actually a very nice read and I learned something new and interesting. I enjoyed thinking about the notion of state equivalence (Definition 1) and how it fits with the loss. I recommend acceptance. Below you'll find some more (hopefully constructive) comments.

The idea of introducing an auxiliary task is reminiscent of the work of Aapo Hyvarinen's group studying identifiability of latent representation in the context of contrastive learning [1] and VAEs [2]. The setup and mathematical results are quite different (they study identifiability of representation up to permutation and rescaling), but I thought the authors might be interested.

I am wondering about the connections between the present work and [3], which is also about self-supervised learning.

Definition 1 states that $\mathcal{O}^2$ is finite, but I don't think it was stated anywhere that $\mathcal{O}$ was finite, it should be stated earlier.

The introduction mentions a finite-data regime, but Theorem 1 assumes the P-JEPA model is well trained, which means the loss is equal to zero for all o and a. This is closer to an infinite data regime IMO, since the loss must be zero on all pairs (o,a). Unless the result applies even if some pairs (o,a) don't have loss equal to zero? If so, it's not clearly explained. Oh I can see the empirical version of the Theorem 1 in the appendix... Please refer to it in the main text.

The concept of "most contracting bisimulation" took me a while to wrap my head around. I would encourage the authors to give textual description of what the set $R^*$ is. My understanding is that it is the set of pairs of observations for which there exists a sequence of actions (which might be length zero) for which executing this sequence of actions for both observations in parallel results in a pair of observations that have different auxiliary value (p(o) = p(o')). Am I close to being right? Also, I'm not sure what the "most contracting" means here...

I thought there was too many negatives in the statement of Theorem 1. B* is itself a complement, and the statement goes: if $(o_i, o_j) \not\in B*$, then $E_\phi(o_i) \not= E_\phi(o_j)$. I would prefer: $(o_i, o_j) \in R*$, then $E_\phi(o_i) \not= E_\phi(o_j)$, which is equivalent and you don't need to define B*. Maybe that's a detail...

Interestingly, the implication in Theorem 1 goes in only one direction, this means that in principle the encoder might not collapse two states even if they are B^* equivalent? Adding these small remarks could be helpful to the reader (I know space is tight, but figure 2 e.g. could be smaller) Oh I can see the experimental section makes that point.

Line 102: "In this work, all plots are obtained when the clusters are the most compact." I don't understand what is meant here.

I'd like to point out that the clustering behavior observed in Figure 2a is not explained by the theory. I think this point could be emphasized more. The authors acknowledged that the theory doesn't predict that bisimilar observations will be collapsed together, but did not explicitly point out that the theory doesn't explain this clustering behavior. The reason why clustering occurs in Figure 2a but not in 2b is left unexplained. This is of course fine, but needs to be acknowledged.

Appendix B: I'm confused about the relationship between Definition 2 (bisimulation) and Definition 3 (Most contracting bisimulation). Can we think of B^* (the most contracting bisimulation) to be the "largest bisimulation", when p = r?

Appendix C: The specification of the dynamics of the environment is vague. Which object is removed? Are the objects moving in space?

**References**

[1] Aapo Hyvarinen, Hiroaki Sasaki, Richard E. Turner. Nonlinear ICA Using Auxiliary Variables and Generalized Contrastive Learning. 2019
[2] Ilyes Khemakhem, Diederik P. Kingma, Ricardo Pio Monti, Aapo Hyvärinen. Variational Autoencoders and Nonlinear ICA: A Unifying Framework.
[3] Julius von Kügelgen, Yash Sharma, Luigi Gresele, Wieland Brendel, Bernhard Schölkopf, Michel Besserve, Francesco Locatello. Self-Supervised Learning with Data Augmentations Provably Isolates Content from Style

**Score:**

4

**Topic Fit:**

3

---

### Official Review · Reviewer_gx28 · 2025-09-11
**Why and How Auxiliary Tasks Improve JEPA Representations**

**Confidence:** 3

**Review:**

**Summary**:

This paper provides theoretical grounding for why auxiliary tasks prevent representation collapse in JEPA models. The core theorem shows that under perfect training in deterministic MDPs, observations with different auxiliary labels or transition dynamics cannot collapse to identical representations.


**Strengths**:
- Clear theoretical contribution linking auxiliary task choice to representational structure
- Solid experimental validation in counting environment matching theory's prediction of 9 distinct clusters
- Practical insight: auxiliary tasks aren't heuristic additions but principled tools for controlling what distinctions the encoder preserves
- Well-designed ablations showing joint training outperforms separate training

**Weaknesses**:

- Theory limited to deterministic MDPs, severely constraining real-world applicability
- Experiments only in highly controlled synthetic environment - unclear how insights transfer to complex visual domains

**Technical Issues**:
- The 69-layer encoder design choice seems arbitrary and potentially confounds results
- Finite data analysis feels disconnected from main narrative

**Overall Assessment**:
Solid theoretical foundation for understanding JEPA+auxiliary architectures with clear practical implications. However, the deterministic assumption and simple experimental validation limit impact. The work establishes useful principles but significant extensions needed for broader applicability.

**Score:**

3

**Topic Fit:**

2

---

### Official Review · Reviewer_fBhL · 2025-09-15
**Review of Submission 114 by Reviewer fBhL**

**Confidence:** 4

**Review:**

**1. Paper Summary**

The paper addresses the problem of representation collapse in Joint-Embedding Predictive Architectures (JEPA).
The authors propose and analyze a simple JEPA variant augmented with an auxiliary regression head.
They present a theoretical result, the "No Unhealthy Representation Collapse" theorem, which guarantees that in deterministic MDPs, training to zero loss prevents non-equivalent observations from collapsing into identical latent representations.
Experiments in a counting environment confirm the theoretical predictions, showing that auxiliary tasks lead to richer, structured embeddings.
As a result, the author states that auxiliary functions provide a principled way to improve JEPA encoders, beyond ad-hoc architectural modifications.


**2. Paper Strenghts**

The paper addresses a well-known problem in representation learning: the tendency of JEPA models to collapse.

The use of "most contracting bisimulation" to formalize the notion of observational equivalence is an interesting way to connect the dynamics and the auxiliary task.
Furthermore, the extension of the proof to a finite-data regime significantly strengthens the practical relevance of the findings.

The experiments are well-designed and provide support for the theory.
The ablation studies, which show that the dynamics loss alone leads to total collapse and the reward loss alone yields only coarse representations, effectively demonstrate that the combination is critical for learning a rich representation.


**3. Paper Weaknesess**

The theory assumes perfect training to zero loss, which is rarely achievable in practice.
The gap between this theoretic formulation and practical training dynamics could have been discussed further.

The theory guarantees that non-equivalent observations will not collapse, but it does not guarantee the inverse: that bisimilar (equivalent) observations will collapse to the same representation.
Achieving this compact representation is often a key goal of self-supervised learning.
However, even the authors note this limitation.


**4. Summary**

The paper's conclusion offers an interesting design principle: improve JEPA encoders by choosing an auxiliary task that captures the phenomenon of interest.
The authors provide theoretical and practical grounding to inspire future discussion on this topic.

**Score:**

4

**Topic Fit:**

2